# Usefulness of Easy-to-Use Risk Scoring Systems Rated in the Emergency Department to Predict Major Adverse Outcomes in Hospitalized COVID-19 Patients

**DOI:** 10.3390/jcm10163657

**Published:** 2021-08-18

**Authors:** Julieta González-Flores, Carlos García-Ávila, Rashidi Springall, Malinalli Brianza-Padilla, Yaneli Juárez-Vicuña, Ricardo Márquez-Velasco, Fausto Sánchez-Muñoz, Martha A. Ballinas-Verdugo, Edna Basilio-Gálvez, Mauricio Castillo-Salazar, Sergio Cásarez-Alvarado, Adrián Hernández-Diazcouder, José L. Sánchez-Gloria, Julio Sandoval, Héctor González-Pacheco, Claudia Tavera-Alonso, Gustavo Rojas-Velasco, Francisco Baranda-Tovar, Luis M. Amezcua-Guerra

**Affiliations:** 1Immunology Department, Instituto Nacional de Cardiología Ignacio Chávez, Tlalpan, Mexico City 14080, Mexico; glezf.julie@gmail.com (J.G.-F.); carlosgar0796@gmail.com (C.G.-Á.); raspringall@yahoo.com (R.S.); maly.brianz@gmail.com (M.B.-P.); yaneli2608@gmail.com (Y.J.-V.); marquezric@hotmail.com (R.M.-V.); fausto22@yahoo.com (F.S.-M.); maalbave_64@yahoo.com (M.A.B.-V.); edbgalvez@gmail.com (E.B.-G.); mau.castilou@gmail.com (M.C.-S.); secazalv@gmail.com (S.C.-A.); adrian.hernandez.diazc@hotmail.com (A.H.-D.); luis_san29@hotmail.com (J.L.S.-G.); sandovalzarate@prodigy.net.mx (J.S.); 2Programa de Maestría en Ciencias Quimicobiológicas, Escuela Nacional de Ciencias Biológicas, Instituto Politécnico Nacional, Miguel Hidalgo, Mexico City 11340, Mexico; 3Pharmacy Faculty, Universidad Autónoma del Estado de Morelos, Cuernavaca 62209, Mexico; 4Coronary Care Unit, Instituto Nacional de Cardiología Ignacio Chávez, Tlalpan, Mexico City 14080, Mexico; hectorglezp@hotmail.com; 5Core Lab, Instituto Nacional de Cardiología Ignacio Chávez, Tlalpan, Mexico City 14080, Mexico; taveramuc@yahoo.com.mx; 6Cardiovascular Intensive Care Unit, Instituto Nacional de Cardiología Ignacio Chávez, Tlalpan, Mexico City 14080, Mexico; gustavorojas08@gmail.com (G.R.-V.); francisco_baranda@yahoo.com.mx (F.B.-T.); 7Health Care Department, Universidad Autónoma Metropolitana–Xochimilco, Coyoacán, Mexico City 04960, Mexico

**Keywords:** COVID-19, diabetes, obesity, inflammation, mechanical ventilation, thrombosis

## Abstract

Background: Several easy-to-use risk scoring systems have been built to identify patients at risk of developing complications associated with COVID-19. However, information about the ability of each score to early predict major adverse outcomes during hospitalization of severe COVID-19 patients is still scarce. Methods: Eight risk scoring systems were rated upon arrival at the Emergency Department, and the occurrence of thrombosis, need for mechanical ventilation, death, and a composite that included all major adverse outcomes were assessed during the hospital stay. The clinical performance of each risk scoring system was evaluated to predict each major outcome. Finally, the diagnostic characteristics of the risk scoring system that showed the best performance for each major outcome were obtained. Results: One hundred and fifty-seven adult patients (55 ± 12 years, 66% men) were assessed at admission to the Emergency Department and included in the study. A total of 96 patients (61%) had at least one major outcome during hospitalization; 32 had thrombosis (20%), 80 required mechanical ventilation (50%), and 52 eventually died (33%). Of all the scores, Obesity and Diabetes (based on a history of comorbid conditions) showed the best performance for predicting mechanical ventilation (area under the ROC curve (AUC), 0.96; positive likelihood ratio (LR+), 23.7), death (AUC, 0.86; LR+, 4.6), and the composite outcome (AUC, 0.89; LR+, 15.6). Meanwhile, the inflammation-based risk scoring system (including leukocyte count, albumin, and C-reactive protein levels) was the best at predicting thrombosis (AUC, 0.63; LR+, 2.0). Conclusions: Both the Obesity and Diabetes score and the inflammation-based risk scoring system appeared to be efficient enough to be integrated into the evaluation of COVID-19 patients upon arrival at the Emergency Department.

## 1. Introduction

Just over a year after the first recorded cases of atypical pneumonia in Wuhan, China, the disease known as coronavirus-19 disease (COVID-19) and officially declared a pandemic by the World Health Organization (WHO) on March 11, 2020, has taken the lives of more than 3.8 million people worldwide [1]. The implementation of certain health measures such as the use of face masks, social distancing, the temporary closure of places of social coexistence, and particularly the advent of vaccination, have given some measure of control over the pandemic. However, this disease continues to pose a threat to global public health.

Within the broad clinical spectrum of COVID-19, most individuals are asymptomatic or present only mild symptoms, although a significant number of patients develop severe disease and ultimately death. Several studies have identified clinical risk factors as well as laboratory and imaging data associated with disease progression, which has led to the design of prognostic scores to give early recognition of those patients with an unfavorable prognosis [2]. Although many of these risk scoring systems are based on clinical and laboratory criteria, it is essential to consider the more affordable techniques in the Emergency Department as an added value for guiding therapeutic strategies in any hospital, including those in low-income countries. Among the most user-friendly scores are those that consider a history of various diseases such as diabetes, obesity, hypertension, and chronic kidney disease especially important [3,4,5,6,7]. On the other hand, as a result of the hyperinflammation underlying severe COVID-19, the efficacy of some calculators that assess the extent of inflammation under other medical conditions is under comprehensive evaluation [8,9,10,11].

Since each clinimetric instrument may perform differently in specific clinical settings, the aim of this study was to compare the ability of several risk scoring systems, rated at the time of patient arrival, to predict the occurrence of major adverse outcomes during hospitalization with COVID-19. The optimal cutoff points (and the intrinsic properties of the diagnostic test) of the risk scoring system that best predicted each major outcome were also evaluated.

## 2. Materials and Methods

### 2.1. Study Participants

The present study was carried out at the National Institute of Cardiology in Mexico City, an academic, tertiary care center devoted to the study and management of cardiovascular diseases and allied conditions. As the COVID-19 pandemic evolved, our hospital converted its Emergency Department and cardiovascular intensive care unit (ICU) to areas dedicated to the critical care of COVID-19 patients. Our hospital received only seriously ill patients during this time, and those with milder forms of the disease were referred to a less-specialized medical center or sent home.

In this retrospective, single-center cohort study, we included data from patients older than 18 years who had been admitted to the Emergency Department from April to August, 2020 with a diagnosis of COVID-19, and met the case definition (probable or confirmed case) of SARS-CoV-2 infection by the WHO [12]. Upon admission, a patient’s disease was classified as moderate or severe: moderate disease was based on clinical signs of pneumonia such as fever, cough, dyspnea or tachypnea; severe disease was defined as the presence of pneumonia and at least one of the following: respiratory rate > 30 breaths/min, severe respiratory distress, or oxygen saturation (SaO_2_) < 90% in room air [13]. At admission, a nasopharyngeal swab was performed, and SARS-CoV-2 positivity was assessed on an RT–PCR test although no serial tests were performed to assess viral clearance. A negative rapid influenza test result was also obtained. Of note, therapies, imaging and laboratory studies, admission to the ICU, and the decision to provide mechanical ventilatory support were performed at the discretion of the treating physicians. In a similar way, the decision to discharge the patient was made solely by the treating physician according to the status of each patient.

The occurrence of each of the major adverse events was assessed from the arrival of each patient to the Emergency Department until discharge or death. Thrombosis was defined as the presence of a blood clot inside an arterial or venous vessel demonstrated by catheterization or by an imaging study: ultrasound, computed tomography, or magnetic resonance imaging. These included acute myocardial infarction, stroke, peripheral artery thrombosis, pulmonary thromboembolism, deep or superficial vein thrombosis, valvular thrombosis, and intracavitary cardiac thrombi. Ventilatory mechanical support was administered at the recommendation of the treating physician and considered as an outcome only when respiratory failure required endotracheal intubation and invasive ventilation. Death was established at the time the patient suffered irreversible cessation of circulatory and respiratory functions or irreversible cessation of all functions of the entire brain, including the brainstem. The composite outcome was constructed by combining the three end points, i.e., the occurrence of either thrombosis, the need for mechanical ventilation, or death.

Upon arrival at the Emergency Department, patients authorized the use of their clinical data for research purposes. This retrospective analysis was approved by the institutional review board (*Comité de Ética en Investigación del Instituto Nacional de Cardiología Ignacio Chávez*) with the protocol number 21-1207. All procedures were carried out in accordance with the 2013 Declaration of Helsinki, its addenda, and local regulations.

### 2.2. Definition of the Risk Scoring Systems Assessed

Clinical and laboratory data were collected from the electronic record by two independent investigators (J.G.-F., C.G.-A.) according to an electronic pro-format file. At this time, all the information necessary to qualify each risk scoring system was obtained. Each database was reviewed by a third investigator (L.M.A.-G.) and discrepancies were solved by reviewing discordant data directly from the medical record. Each of the risk scoring systems and how they must be rated are described in Appendix A.

*Charlson comorbidity index*. A validated, simple, and readily applicable method to estimate the impact of comorbid disease for survival, especially in intensive care health services [3]. The Charlson comorbidity index scores variables such as age, cardiovascular disease, lung disease, liver disease, dementia, diabetes, AIDS, and neoplasms. The most recent version already considers if the patient has COVID-19 although this is for research purposes only and this item does not (yet) affect the results. In COVID-19 patients, the per-point increase in the Charlson comorbidity index score increases the risk of mortality by 16% [14]. There is an online calculator that performs the algorithm in an automated way (https://www.mdcalc.com/charlson-comorbidity-index-cci#creator-insights (accessed on 25 April 2021)).

*LOW-HARM score*. An easy-to-measure tool to predict mortality in hospitalized COVID-19 patients, specifically designed for clinical settings without access to inflammation markers to test COVID-19, such as C-reactive protein, D-dimer, or interleukin-6. The acronym LOW-HARM represents each of the parameters that make up the score, namely lymphopenia, oxygen saturation, white blood cells, hypertension, age, renal injury, and myocardial injury. Each of the parameters dichotomously scores as present/absent according to pre-established cutoff points, except for age, which scores differently according to the decade of age to which the patient corresponds. According to the optimal cutoff point, the LOW-HARM score may reach 63% sensitivity and 97.5% specificity to predict death [4]. Online calculator: https://lowharmcalc.com/ (accessed on 26 April 2021).

*CALL score* is a modeled clinical tool to predict the progression of COVID-19 in hospitalized patients. Its name is the acronym for the set of criteria that are considered for its evaluation: comorbidities, age, lymphocyte count, and lactic dehydrogenase. Each criterion provides a different score, depending on the weight of that variable. Using a cutoff point of 6, the CALL score shows a positive predictive value of 50.7% and a negative predictive value of 98.5% for mortality in COVID-19 [5]. Online calculator: https://www.rccc.eu/COVID/CALL.html (accessed on 25 April 2021).

*Obesity and Diabetes score*. A validated scoring system to predict death from COVID-19. In its evaluation, the presence of pneumonia, diabetes with emphasis on early onset (age < 40 years), age, chronic kidney disease, immunosuppression, chronic obstructive pulmonary disease, and obesity are considered. Each condition provides a different value. The Obesity and Diabetes score has a high ability to discriminate lethality, with a C-statistic of 0.83 [6].

*PH-Covid19 scoring system*. A system developed to predict mortality in COVID-19, based on data from the medical history, which comprises age, sex, diabetes, chronic obstructive pulmonary disease, immunosuppression, hypertension, obesity, and chronic kidney disease. The final score is used to stratify patients into four categories of risk of death [7].

*Inflammation-based risk scoring system*. An instrument originally developed to predict mortality in the setting of acute coronary syndrome, which assesses the state of systemic inflammation. In its evaluation, the serum levels of albumin and high sensitivity C-reactive protein are considered, as well as the white blood cell count, according to different pre-specified cutoff points that allow stratifying the degree of inflammation into three categories [9]. The inflammation-based risk scoring system has already been shown to be useful in predicting the need for mechanical ventilation in hospitalized patients with severe COVID-19 [15].

*Neutrophil-lymphocyte ratio (NLR)*. A simple formulation developed to assess changes in leukocyte subpopulations in response to systemic inflammation, which has been used extensively in a variety of medical conditions. The NLR results from dividing absolute neutrophil count/absolute lymphocyte count [10]. In the setting of COVID-19, the NLR has been shown to be useful for predicting disease progression to a critical stage (C-statistic = 0.84) [16].

*HScore*. This score is used to estimate the individual’s risk for the reactive hemophagocytic syndrome, a hyperinflammatory condition caused by dysregulated immune responses. The following items are included: immunosuppression, body temperature, organomegaly, cytopenias, ferritin, triglycerides, fibrinogen, aspartate aminotransferase, and features of hemophagocytosis in bone marrow aspirate. Different cutoff points are assigned to each criterion. Thus, the probability of having reactive hemophagocytic syndrome is determined [11]. In the case of COVID-19, the use of the HScore was initially proposed to guide therapeutic strategies; however, evidence suggests marginal utility at best [17]. Online calculator: https://www.mdcalc.com/hscore-reactive-hemophagocytic-syndrome (accessed on 25 April 2021).

### 2.3. Statistical Analysis

Frequencies and percentages were used to describe categorical variables, while a mean with ±1 standard deviation (SD) or a median with an interquartile range (IQR) were used to describe numerical variables as appropriate. Each of the risk scoring systems was tested for its ability to predict the in-hospital occurrence of the following outcomes: vascular thrombosis, need for mechanical ventilation, death, and the composite outcome. These analyses were performed using the area under the receiver operating characteristic (ROC) curve (AUC) with 95% confidence intervals (95% CI). The risk scoring system that showed the best performance for predicting each of the outcomes was subsequently analyzed to identify its optimal cutoff point, according to the Youden’s J statistic (J = sensitivity + specificity − 1). Finally, this cutoff point was used to identify the intrinsic properties of the diagnostic test, in particular sensitivity, specificity, positive predictive value (PPV), negative predictive value (NPV), positive likelihood ratio (LR+), and negative likelihood ratio (LR-).

In a final set of analyses, cumulative survival curves for the occurrence of each major adverse outcome during the hospital stay were constructed using the Kaplan–Meier method, and differences were assessed using the log-rank test. Only the risk scoring system that showed the best performance was evaluated. The length of survival was defined as an entry into the study (i.e., admission to the Emergency Department) until the appearance of each outcome or discharge from the hospital. Discharged patients were considered censored observations at the time of the last day of hospital stay. The Kaplan–Meier curves were truncated until the last patient had a main outcome or was discharged. A maximum time frame of 90 days was used.

All analyses were two-sided and a value of *p* < 0.05 was established for significance. For the calculations, GraphPad Prism v.9 (GraphPad Software, La Jolla, CA (USA) and MedCalc v.19 (MedCalc Software, Mariakerke (Belgium) statistical software, as well as the online calculator Fisterra (https://www.fisterra.com/mbe/ (accessed on 25 April 2021)) were used.

## 3. Results

During the study period, a total of 157 patients (66.3% male) were recruited having a mean age of 55 ± 12 years. The main clinical and demographic characteristics are summarized in Table 1. A significant number of comorbidities were observed, hypertension (46.4%), diabetes (36.3%), obesity (29.2%), and dyslipidemia (13.3%). Different end-stage organ diseases were also frequently found, and the median Charlson comorbidity index was 2 (IQR, 1 to 4). An average delay of 7 days was found from the onset of symptoms to arrival at the Emergency Department. The main clinical and laboratory characteristics at hospital admission are shown in Table 2. One-third of the patients had fever, while the mean respiratory rate was 26.0 ± 11.8 breaths/min, with mean oxygen saturation (at room air) of 79.5 ± 13.1%. Almost all patients (79.6%) were classified as having severe COVID-19; the rest had moderate disease. An important inflammatory response was found, which was characterized by a marked elevation of C-reactive protein, ferritin, and interleukin-6, while the biomarkers of cell damage (troponin I) and fibrinolysis (D-dimer) were also markedly elevated.

Depending on the degree of respiratory failure, patients sequentially received supplemental oxygen through nasal cannula, high-flow oxygen therapy, non-invasive ventilation, and finally endotracheal intubation with mechanical ventilatory support. The prone position was part of the standard medical treatment. In addition, a total of 139 patients (88.5%) received heparins, 54 (34.3%) received dexamethasone, 21 (13.3%) received tocilizumab (an interleukin-6 inhibitor), and 11 (7.0%) received Jak-STAT inhibitors (baricitinib or ruxolitinib). No patient received convalescent plasma. At the time of the study, there were no COVID-19 vaccines or remdesivir in clinical use and the typing of viral variants was not yet available.

A total of three in-hospital outcomes were studied, namely the occurrence of vascular thrombosis, the need for mechanical ventilation, and death. In addition, to increase statistical efficiency, the occurrence of a composite outcome was analyzed, in which the three end points were combined as a primary outcome measure. Of the 157 patients with hospital follow-up, 32 (20.3%) had at least one thrombotic/thromboembolic event, as follows: deep vein thrombosis (20 cases), pulmonary thromboembolism (6 cases), acute myocardial infarction (7 cases), stroke (2 cases) and intracardiac/valvular thrombosis (2 cases). Furthermore, 80 (50.9%) patients required mechanical ventilation and 52 (33.1%) eventually died. Finally, a composite outcome was found in 96 (61.1%) patients.

The analysis on the ability of each risk scoring system to predict major outcomes is summarized in Table 3. Scores based primarily on the history of comorbid conditions adequately predicted the need for mechanical ventilation and occurrence of death but were unable to identify patients who would develop thrombosis. On the other hand, scores based on inflammation (except for the HScore) efficiently identified the occurrence of any of the main outcomes evaluated in this study. Specifically, the inflammation-based risk scoring system showed the greatest ability to predict thrombosis (AUC, 0.63; 95% CI: 0.52 to 0.74; *p* = 0.020), although its performance was far from optimal, while the Obesity and Diabetes score was the best for predicting the need for mechanical ventilation (AUC, 0.96; 95% CI: 0.93 to 0.99; *p* < 0.001), death (AUC, 0.86; 95% CI: 0.79 to 0.92; *p* < 0.001) and composite outcome (AUC, 0.89; 95% CI: 0.84 to 0.94; *p* < 0.001). A comparative ROC analysis of the risk scoring systems that performed best in predicting each clinical outcome is presented in Figure 1. The inflammation-based risk scoring system was not significantly better (*p* = 0.737) than the NLR for predicting vascular thrombosis. On the other hand, the Obesity and Diabetes score was better than any of the other risk scoring systems for predicting the need for mechanical ventilation (*p* < 0.0001 for all comparisons), death (*p* < 0.001 for all comparisons), and composite outcome (*p* < 0.01 for all comparisons).

The attributes of this diagnostic test of the risk scoring system that predict each major outcome the best are presented in detail in Table 4. Briefly, at a cutoff of ≥5 points (according to Youden’s J statistic), the inflammation-based risk scoring system at the time of hospital admission provided an LR+ rating of 2.0 (95% CI: 1.3 to 3.0) for the development of thrombosis during hospitalization. The Obesity and Diabetes score (at a cutoff of ≥5 points) provided an impressive LR+ of 23.7 (95% CI: 7.8 to 72.1) for the need of mechanical ventilation, and a LR+ of 15.6 (95% CI: 5.1 to 47.5) for the occurrence of the composite outcome. At a cutoff of ≥7 points, it provided a LR+ of 4.6 (95% CI: 2.9 to 7.1) for in-hospital mortality.

Once the usefulness of the inflammation-based risk scoring system and the Obesity and Diabetes score for predicting major outcomes was identified, we estimated the percentage of COVID-19 patients who remained free of each outcome during hospitalization. As noted in Figure 2, patients with ≥5 points in the inflammation-based risk scoring system had a hazard ratio (HR) for thrombosis of 1.5 (95% CI: 0.6 to 3.7; *p* = 0.359). In contrast, patients with ≥5 points in the Obesity and Diabetes score had an HR = 36.0 (95% CI: 21.4 to 60.7; *p* < 0.0001) for mechanical ventilation and an HR = 3.0 (95% CI: 1.9 to 4.6; *p* < 0.0001) for the composite outcome. Patients with an Obesity and Diabetes score ≥7 points had an HR = 3.3 (95% CI: 1.9 to 5.9; *p* = 0.0006) for overall mortality.

## 4. Discussion

In this study, the ability of different risk scores to predict major adverse outcomes in hospitalized patients with COVID-19 was evaluated. Our results show that the Obesity and Diabetes score, one of the easiest scores to use since it is based exclusively on data from the medical history, had the best predictive ability for most major outcomes, especially the need for mechanical ventilation. On the other hand, the inflammation-based risk scoring system, a simplified score comprising three readily available inflammatory biomarkers, shows the highest ability to predict the occurrence of thrombosis.

The main finding of this study was the exceptional performance of the Obesity and Diabetes score in the early prediction of the need for mechanical ventilation and overall mortality in patients with COVID-19. Although this score is made up of several items, there are some that are of special importance. One of the most relevant criteria is the history of diabetes, particularly early-onset diabetes, which is a phenotype the prevalence of which is alarmingly increasing. Diabetes has an underlying chronic inflammatory process with the consequent dysfunctional immune response, which makes diabetic patients more susceptible to infections [18]. In the context of SARS-CoV-2 infection, diabetes has been consistently associated with disease progression, increased requirement for mechanical ventilation, longer hospital stay, and a marked increase in mortality [19,20,21,22]. Likewise, in diabetic patients with COVID-19, it has been observed that advanced age, the use of insulin for glycemic control, and sustained hyperglycemia contributed to an even worse prognosis [23,24]. Another metabolic condition considered in the Obesity and Diabetes score was obesity, a highly prevalent disease in which excessive accumulation of adipose tissue, especially visceral adipose tissue, interferes with homeostasis by stimulating the release of inflammatory mediators such as interleukin-6, tumor necrosis factor, and other adipokines [25,26]. In COVID-19, obesity has been widely recognized as one of the main predictors of poor prognosis. A retrospective cohort showed a gradual increase in the need for mechanical ventilation as the body mass index increases, reaching a 7 times higher risk in those patients with a body mass index > 35 [27]. A systematic review found that obesity was a significant risk factor for ICU admission (odds ratio (OR) 1.21) as well as for mechanical ventilation (OR 2.05) [28]. Other studies already confirmed obesity as among the three most important predictors of severe COVID-19 [29]. It is highly relevant in the current era, where confinement has led to less physical activity and marked changes in diet, which inevitably result in weight gain, thus acquiring a potential risk factor for serious disease in the event of contracting SARS-CoV-2 infection [30]. 

Chronic kidney disease is another condition of persistent and unregulated inflammation that is also included in the Obesity and Diabetes score. A recent meta-analysis that studied risk factors for severe disease and death in COVID-19 patients found that chronic kidney disease confers an OR of 3.5 for severe disease and 5.3 for death, ranking it as the second-most important risk factor for death, behind only advanced age (>65 years) [31]. The presence of comorbid conditions in other viral respiratory diseases such as influenza has been shown to have a negative effect during infection, although to a lesser extent than in COVID-19. A nationwide retrospective study compared patients hospitalized for COVID-19 from 1 March to 30 April 2020, and patients hospitalized for influenza in France from 1 December 2018, to 28 February 2019. Patients with COVID-19 were more obese and had diabetes and hypertension more often than patients with influenza. Hospitalized COVID-19 patients developed acute respiratory failure, pulmonary embolism, septic shock, and stroke more frequently than their counterparts with influenza, while in-hospital mortality was higher in patients with COVID-19 (16.9 vs. 5.8%) [32].

The other major finding in our study was the usefulness of the inflammation-based risk scoring system for predicting all adverse outcomes although this score was particularly efficient for vascular thrombosis. (A thrombosis is a clinical entity in which an inflammatory component has been widely recognized as part of its pathogenesis [33]). Each biomarker considered in the inflammation-based risk scoring system was independently assessed for thrombosis in patients with COVID-19. Indeed, C-reactive protein was independently associated with the presence of a deep-vein thrombosis demonstrated by duplex ultrasound as well as a pulmonary embolism demonstrated by computed tomography pulmonary angiography [34,35]. Moreover, C-reactive protein levels may predict the occurrence of venous thromboembolism in critically ill COVID-19 patients admitted to the ICU with an AUC of 0.75; for comparison, the AUC of the D-dimer in this study was only 0.64 [36]. In COVID-19 patients receiving prophylactic anticoagulation, elevated levels of C-reactive protein (OR 2.7) and D-dimer (OR 6.7), as well as thrombocytosis (OR 3.5) at admission were predictive of coagulation-associated complications during hospitalization [37]. Decreased serum albumin levels (<3.5 g/dL) were found more frequently in COVID-19 patients with thromboembolic events compared to thrombosis-free patients, usually in association with elevated levels of C-reactive protein and D-dimer [38]. Parallel to its response as an acute phase reactant, hypoalbuminemia in COVID-19 may reflect a dysregulated immune response in the early stages of the disease, which could lead to increased capillary permeability and the release of albumin into the interstitium. Interestingly, albumin supplementation in COVID-19 patients produced a reduction in D-dimer levels and a decrease in overall mortality with a trend toward fewer thromboembolic events [39]. In a recent meta-analysis, leukocytosis (with lymphopenia) and elevated D-dimer levels were the major contributors for the occurrence of deep venous thrombosis in COVID-19 [40]. A process of immunothrombosis triggered by neutrophil extracellular traps (NETs) has recently been described in COVID-19. Indeed, myeloperoxidase (MPO)-DNA complexes, which reflect active NET formation, were increased in the plasma of patients with acute respiratory distress syndrome, and lung biopsies have shown NET-containing microthrombi with infiltration of neutrophils and platelets [41]. NET-related immunothrombosis in COVID-19 is a process dependent on complement activation, which opens a mechanistic possibility of therapeutic intervention [42]. An interesting study found that plasma levels of NET markers are associated with the need for ventilation and short-term mortality in patients with moderate-to-severe COVID-19; all NET markers declined four months after infection [43]. Recently, the existence of an immune-mediated coagulopathy, with induction of autoantibodies (mainly antiphospholipid antibodies) parallel to hyperinflammation has been suggested [44,45].

We are aware that our study has limitations. First, we included mostly critically ill patients admitted to a single, highly specialized cardiovascular center, and the results could be influenced by the nature of our institute (and our country), where hypertension, diabetes, and obesity are unusually frequent [46]. Therefore, the exceptional performance of the Obesity and Diabetes score may not be applicable to other populations. However, it should be noted that large geographic regions other than North America are experiencing an emergency of noncommunicable diseases, which are responsible for a total of 71% of all deaths globally [47]. Second, the lack of bone marrow aspirate could be the basis for the particularly poor performance of the HScore in predicting hyperinflammation in our COVID-19 patients as they did not undergo such a procedure. However, studies have already shown the lack of usefulness of the HScore in the context of COVID-19 [17,48]. Third, although this study may be considered as an external validation analysis of each risk scoring system in COVID-19, it is necessary to assess its usefulness in patients with milder forms of the disease, as well as among patients with different ethnic and geographic backgrounds. Finally, the arrival of highly effective vaccination schedules and the appearance of new viral variants of concern pose a new challenge to evaluate the efficacy of these risk scoring systems in each of the specific groups of patients. Additionally, since the use of these clinimetric tools categorizes patients at high risk of adverse outcomes, it is conceivable that clinical management can also be modified according to risk levels. However, this is purely speculative and further studies are required to assess the impact of risk scoring systems on the treatment of COVID-19 patients.

## 5. Conclusions

Although deaths from COVID-19 are declining in some regions due to vaccination, pandemic activity is higher than ever worldwide. For this reason, the use of easy-to-use clinimetric tools that allow early recognition of high-risk patients remains of utmost importance. Among the multiple instruments available to classify the severity of patients with COVID-19, we found that two of the simplest scores have an excellent performance in predicting the occurrence of the main adverse outcomes during hospitalization.

## Figures and Tables

**Figure 1 jcm-10-03657-f001:**
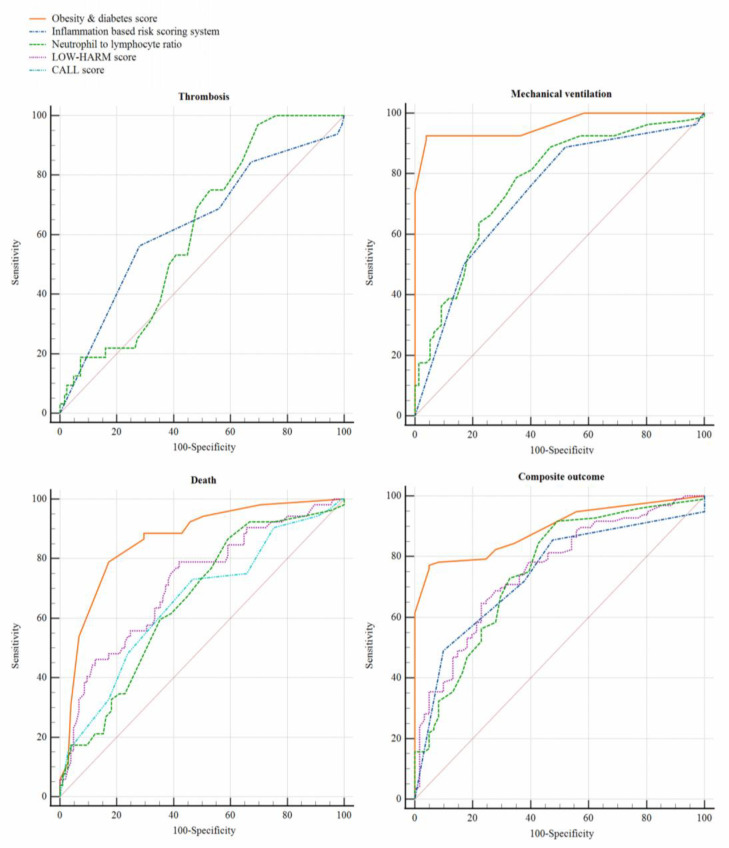
A comparative ROC analysis of the risk scoring systems that significantly discriminated the occurrence of each major adverse outcome. The inflammation-based risk scoring system was not better than the neutrophil-lymphocyte ratio (*p* = 0.737) in predicting thrombosis. In contrast, the Obesity and Diabetes score was better than any of the other risk scoring systems in predicting the need for mechanical ventilation (*p* < 0.0001 for all comparisons), death (*p* < 0.001 for all comparisons), and the composite outcome (*p* < 0.01 for all comparisons).

**Figure 2 jcm-10-03657-f002:**
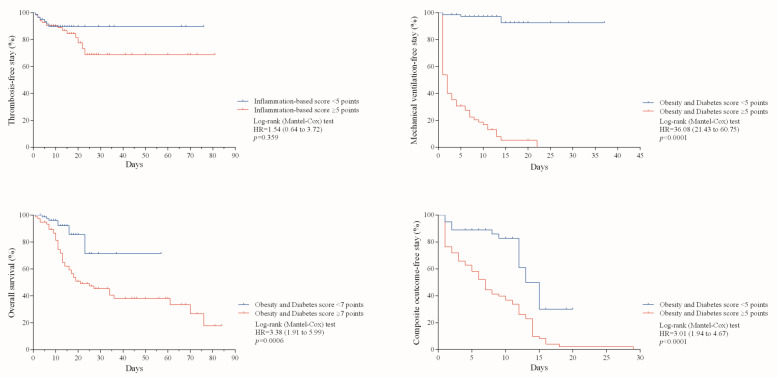
Cumulative survival curves for the occurrence of major adverse outcomes during hospital stay.

**Table 1 jcm-10-03657-t001:** Demographic and clinical data of study participants.

	COVID-19 Patients (*n* = 157)
Age in years, mean ± SD	55 ± 12
Male sex, *n* (%)	105 (66.3)
Body mass index ≥ 30 kg/m^2^, *n* (%)	46 (29.2)
Current smoking, *n* (%)	30 (19.1)
Coexisting conditions, *n* (%)	
Diabetes mellitus	57 (36.3)
Hypertension	73 (46.4)
Dyslipidemia	21 (13.3)
Coronary artery disease	14 (8.9)
Stroke	5 (3.1)
Chronic heart failure	9 (5.7)
Chronic kidney disease	19 (12.1)
Chronic obstructive pulmonary disease	7 (4.4)
Autoimmune diseases	9 (5.7)
Organ transplant	7 (4.4)
Cancer	4 (2.5)
Charlson comorbidity index, median (IQR)	2 (1 to 4)

**Table 2 jcm-10-03657-t002:** Main findings at hospital admission of study participants.

	COVID-19 Patients (*n* = 157)
Days of symptom onset, median (IQR)	7.0 (4.7 to 9.0)
*Clinical data*	
Temperature > 37.3 °C, *n* (%)	57 (36.3)
Respiratory rate, breaths/min	26.0 ± 11.8
Heart rate, beats/min	96.2 ± 19.8
Systolic blood pressure, mmHg	124.7 ± 21.0
Diastolic blood pressure, mmHg	75.9 ± 13.5
Oxygen saturation at room air, %	79.5 ± 13.1
Classified as severe COVID-19, *n* (%)	125 (79.6)
*Laboratory values*	
White cell count (×103 per mm^3^), median (IQR)	8.9 (6.1 to 12.3)
Neutrophils (×103 per mm^3^), median (IQR)	7.7 (4.8 to 11.0)
Lymphocytes (×103 per mm^3^), median (IQR)	0.8 (0.6 to 1.1)
Platelets (×103 per mm^3^), median (IQR)	207 (164 to 275)
Hemoglobin, g/dL, median (IQR)	14.7 (13.2 to 16.0)
Albumin, g/dL, median (IQR)	3.4 (3.1 to 3.8)
Serum creatinine, mg/dL, median (IQR)	1.0 (0.8 to 1.4)
Troponin I, ng/mL, median (IQR)	12.8 (6.1 to 63.8)
Creatine kinase, U/L, median (IQR)	105 (49 to 199)
D-dimer, ng/mL, median (IQR)	390 (228 to 666)
Fibrinogen, mg/dL, median (IQR)	5.3 (4.4 to 6.1)
C-reactive protein, mg/L, median (IQR)	145 (61 to 256)
Ferritin, μg/L, median (IQR)	590 (270 to 1101)
Interleukin 6, pg/mL, median (IQR)	14.9 (4.5 to 73.5)

Data are presented as mean ± standard deviation unless otherwise specified.

**Table 3 jcm-10-03657-t003:** The area under the receiver operating characteristic curve (AUC-ROC) of each risk scoring system to predict major outcomes in COVID-19 patients.

Risk Scoring System	Thrombosis (*n* = 32)	Mechanical Ventilation (*n* = 80)	Death (*n* = 52)	Composite Outcome (*n* = 96)
Charlson comorbidity index	0.52 (0.41 to 0.63)	0.50 (0.40 to 0.59)	0.60 (0.51 to 0.70)	0.52 (0.43 to 0.61)
LOW-HARM score	0.58 (0.47 to 0.68)	0.72 (0.64 to 0.80)	0.71 (0.63 to 0.80)	0.75 (0.67 to 0.83)
CALL score	0.52 (0.41 to 0.63)	0.61 (0.52 to 0.70)	0.65 (0.56 to 0.74)	0.60 (0.51 to 0.69)
Obesity and Diabetes score	0.59 (0.48 to 0.70)	0.96 (0.93 to 0.99)	0.86 (0.79 to 0.92)	0.89 (0.84 to 0.94)
PH-Covid19 score	0.52 (0.40 to 0.63)	0.56 (0.47 to 0.65)	0.64 (0.55 to 0.73)	0.59 (0.50 to 0.68)
Inflammation-based risk scoring system	0.63 (0.52 to 0.74)	0.73 (0.65 to 0.81)	0.60 (0.51 to 0.70)	0.74 (0.66 to 0.82)
Neutrophil-lymphocyte ratio	0.61 (0.51 to 0.70)	0.76 (0.69 to 0.84)	0.65 (0.56 to 0.74)	0.75 (0.67 to 0.83)
HScore	0.54 (0.43 to 0.64)	0.53 (0.44 to 0.62)	0.53 (0.43 to 0.62)	0.55 (0.46 to 0.64)

**Table 4 jcm-10-03657-t004:** Attributes as a diagnostic test of the risk scoring system that worked best to predict each major outcome.

	Thrombosis	Mechanical Ventilation	Death	Composite Outcome
Scoring system (optimal cutoff point by the Youden’s index)	Inflammation-based risk scoring system (≥5 points)	Obesity and Diabetes score (≥5 points)	Obesity and Diabetes score (≥7 points)	Obesity and Diabetes score (≥5 points)
Sensitivity	56.2% (37.8 to 73.1)	92.5% (83.8 to 96.9)	78.8% (64.9 to 88.4)	77.0% (67.1 to 84.7)
Specificity	72.0% (63.1 to 79.4)	96.1% (88.2 to 98.9)	82.8% (73.9 to 89.2)	95.0% (85.4 to 98.7)
Positive predictive value	33.9% (21.8 to 48.3)	96.1% (88.2 to 98.9)	69.4% (55.9 to 80.4)	96.1% (88.2 to 98.9)
Negative predictive value	86.5% (78.1 to 92.1)	92.5% (83.8 to 96.9)	88.7% (80.4 to 93.9)	72.5% (61.2 to 81.6)
Positive likelihood ratio	2.0 (1.3 to 3.0)	23.7 (7.8 to 72.1)	4.6 (2.9 to 7.1)	15.6 (5.1 to 47.5)
Negative likelihood ratio	0.6 (0.4 to 0.9)	0.08 (0.04 to 0.17)	0.2 (0.1 to 0.4)	0.2 (0.1 to 0.3)

## Data Availability

The data presented in this study are available on request from the corresponding author. The data are not publicly available due to restrictions associated with the protection of personal data in force in Mexico.

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
