# Peer review of "Usefulness of Easy-to-Use Risk Scoring Systems Rated in the Emergency Department to Predict Major Adverse Outcomes in Hospitalized COVID-19 Patients"

_jcm, 2021, doi:10.3390/jcm10163657_

Round 1
Reviewer 1 Report
Thank you for inviting me to review the article entitled "Usefulness of easy-to-use risk scoring systems rated in the 3 Emergency Department to predict major adverse outcomes in 4 hospitalized COVID-19 patients" (jcm-1282772). The study was aimed to compare the ability of several risk scoring to predict the occurrence of major adverse outcomes during hospitalization in patients with COVID-19. My comments are as follows
1. The study was conducted in a single center that mainly received critical patients. Hence, the selection bias of using these scoring systems for the group of critical patients, rather than general patients, should be clearly addressed. The authors should also explain that the result of this study might only apply to this specific group of patients, which could largely limit the clinical application.
2. The classification of moderate or severe patients in the section of Methods is confusing. Since this study included both moderate or severe patients as their target group, using a sole definition for their inclusion criteria is more understandable for readers.
3. The major adverse outcomes during hospitalization should be defined in the section of Methods.
4. Did the authors exclude patients that were children, pregnant, trauma, with valid DNAR orders, with missing data, etc.?
5. The authors used 8 different scoring systems to examine their predictive values for major adverse events. The authors may use a supplementary table to enlist how these scoring systems are calculated and which kinds of data were needed. The population that these scoring systems were originally applied to should also be identified since this study was conducted in a specific group of hospitalized patients.
Author Response
- The study was conducted in a single center that mainly received critical patients. Hence, the selection bias of using these scoring systems for the group of critical patients, rather than general patients, should be clearly addressed. The authors should also explain that the result of this study might only apply to this specific group of patients, which could largely limit the clinical application.
R= We agree with the Reviewer's suggestion. Consequently, we have prioritized this as the main limitation of our study. Additionally, we add some considerations about the clinical limitations of our study. In this version, the limitations section appears as follows: “First, we included mostly critically ill patients admitted to a single, highly specialized cardiovascular center, and the results could be influenced by the nature of our institute (and our country) where hypertension, diabetes, and obesity are unusually frequent. Therefore, the exceptional performance of the Obesity & Diabetes score may not be applicable to other populations…. Finally, the arrival of highly effective vaccination schedules and the appearance of new viral variants of concern pose a new challenge to evaluate the efficacy of these risk scoring systems in each of the specific groups of patients….”
- The classification of moderate or severe patients in the section of Methods is confusing. Since this study included both moderate or severe patients as their target group, using a sole definition for their inclusion criteria is more understandable for readers.
R= Indeed, the definition of patients is confusing. We have modified the definitions to make them more understandable to readers, although we prefer to keep the separation between moderate and severe disease, as this is how the severity of patients is described in the table and text. Now, the definition is as follows: “At hospital admission, patients were classified to have moderate disease on the basis of clinical signs of pneumonia such as fever, cough, dyspnea, and/or tachypnea; severe disease was defined as the presence of pneumonia and at least one of the following: respiratory rate >30 breaths/min, severe respiratory distress, or oxygen saturation (SaO2) <90% on room air”. Of note, this is the definition used by the World Health Organization.
- The major adverse outcomes during hospitalization should be defined in the section of Methods.
R= Our mistake. The following paragraph was included defining the major adverse outcomes: “The occurrence of each of the major adverse events was assessed from the arrival of each patient to the Emergency Department until their hospital discharge or death. Thrombosis was defined as the presence of a blood clot inside an arterial or venous vessel, demonstrated by catheterization or by an imaging study including ultrasound, computed tomography, or magnetic resonance imaging. These included acute myocardial infarction, stroke, peripheral artery thrombosis, pulmonary thromboembolism, deep or superficial vein thrombosis, valvular thrombosis, and intracavitary cardiac thrombi. Ventilatory mechanical support was administered at the consideration of the treating physician and was considered as an outcome only when respiratory failure required endo-tracheal intubation and invasive ventilation. Death was established at the time a patient suffered the irreversible cessation of circulatory and respiratory functions, or the irreversible cessation of all functions of the entire brain, including the brainstem. The composite outcome was constructed by combining the three end points, i.e., the occurrence of either thrombosis, the need for mechanical ventilation, or death.”
- Did the authors exclude patients that were children, pregnant, trauma, with valid DNAR orders, with missing data, etc.?
R= This study was conducted in individuals older than 18 years; no other exclusion criteria were used. Now we have specified both in the Abstract and in the Methods that it was carried out in adult population.
- The authors used 8 different scoring systems to examine their predictive values for major adverse events. The authors may use a supplementary table to enlist how these scoring systems are calculated and which kinds of data were needed. The population that these scoring systems were originally applied to should also be identified since this study was conducted in a specific group of hospitalized patients.
R= A Table (in the Appendix) has already been included that summarizes the type of data that each risk scoring system requires. In addition, links to online calculators are included to make it easier for readers.
- The clinical application or feasibility of using these scoring systems, as the results of this study, might be addressed in the section of Discussion. That is, should a patient be categorized as high risk by these scoring systems, the clinical management would have modified accordingly to better patient outcomes.
R= This is a very interesting subject, although it is also speculative. We have included some lines at the end of the Discussion on this matter. “…Additionally, since the use of these clinimetric tools categorizes patients at high risk of adverse outcomes, it is conceivable that clinical management can also be modified ac-cording to risk levels. However, this is purely speculative and further studies are required to assess the impact of risk scoring systems on the treatment of COVID-19 patients.”
The authors are indebted to the Reviewer, we are sure that their comments have substantially improved the quality of our manuscript.
Reviewer 2 Report
Congratulations on this work of interest that could lead to improved triage policies in hospital for COVID patients.
I have nevertherless some points to raise :
1. It would be good to specify a little better the population characteristics :
- The number of days of mechanical ventilation
- The use of NIV, high flow oxygen therapy, prone position, ...
- The use of anti-inflammatory therapy or anticoagulation
- Type of COVID Variant
(...)
2. There is no comparative approach between the different scores and their AUC for each major outcome. It would have been good to see if there were differences between the scores.
3. The study (REF 6) seems not to be really consistent with your results:
For peole with a score between 8 to >12, survival probality is between 55 and 70% while it is 40% in your study.
How do you explain such result ?
4. Maybe a colorimetric figure should be more easy to read than the Table 3.
Author Response
REVIEWER 2
Congratulations on this work of interest that could lead to improved triage policies in hospital for COVID patients.
I have nevertherless some points to raise :
- It would be good to specify a little better the population characteristics :
- The number of days of mechanical ventilation
- The use of NIV, high flow oxygen therapy, prone position, ...
- The use of anti-inflammatory therapy or anticoagulation
- Type of COVID Variant
R= We agree with the Reviewer's observation. We have included the following paragraph in the Results section, where we better describe the characteristics of the study population: “Depending on the degree of respiratory failure, patients sequentially received supplemental oxygen through nasal cannula, high-flow oxygen therapy, non-invasive ventilation, and finally endotracheal intubation with mechanical ventilatory support. The prone position was part of the standard medical treatment. In addition, a total of 139 patients (88.5%) received heparins, 54 (34.3%) received dexamethasone, 21 (13.3%) received tocilizumab (an interleukin-6 inhibitor), and 11 (7.0%) received Jak-STAT inhibitors (baricitinib or ruxolitinib). No patient received convalescent plasma. At the time of the study, there were no COVID-19 vaccines or remdesivir in clinical use and the typing of viral variants was not yet available.”
- There is no comparative approach between the different scores and their AUC for each major outcome. It would have been good to see if there were differences between the scores.
R= An important omission in our study. Please find the analysis you are requesting in Figure 1 (a new figure). In addition, the following paragraph was included in the Results: ”A comparative ROC analysis of the risk scoring systems that performed best in predicting each clinical outcome is presented in Figure 1. The inflammation-based risk scoring system was not significantly better (P=0.737) than the NLR in predicting vascular thrombosis. Contrarily, the Obesity & Diabetes score was better than any of the other risk scoring systems in predicting the need for mechanical ventilation (P<0.0001 for all comparisons), death (P<0.001 for all comparisons), and the composite outcome (P<0.01 for all comparisons).”
- The study (REF 6) seems not to be really consistent with your results:
For peole with a score between 8 to >12, survival probality is between 55 and 70% while it is 40% in your study.
How do you explain such result ?
R= The study by Bello-Chavolla et al was developed on the data from the General Directorate of Epidemiology of the Mexican Ministry of Health, including a total of 51,633 confirmed cases of COVID-19. This study included anyone who had tested positive for SARS-CoV-2, regardless of their age and with any level of COVID-19 severity. That is, it included both children and asymptomatic or mildly ill individuals. Likewise, outpatients and hospitalized patients were included equally. In contrast, our study included only adults who required hospitalization for moderate to severe illness.
These differences underlie the 55-70% survival observed in the Bello-Chavolla study, which contrasts with the 40% found in our study (Obesity & Diabetes score 8-12).
- Maybe a colorimetric figure should be more easy to read than the Table 3.
R= Although we agree with the Reviewer, we prefer to use a colorimetric system in Figure 1, which contrasts the different risk scoring systems, rather than in Table 3. We hope this satisfies the Reviewer’s concern.
The authors are indebted to the Reviewer, we are sure that their comments have substantially improved the quality of our manuscript.
Reviewer 3 Report
Please, find the attachment.
[Overall opinion]
This helps identify patients who need prompt treatment. But in order to generalize the author’s ideas, the validation should be conducted. The validation can be conducted with external data or internal data.
The validation with external data is not available in many cases because it is difficult to collect additional data. In that case authors can proceed with internal validation to claim the scoring system as a prediction tool. It is easily found the internal validation in Google. It is difficult to recognize the scoring system without validation results.
[For Table 3]
1) It is recommended to remove p-values on Table 3. Because the significant p-value obtained from ROC analysis means that AUC is significantly different from 0.5. Thus, the p-value less than 0.05 on ROC curve dose not have meaning. Instead of that, the amount of AUC value should be used. Providing the p-values from AUC can make readers confused. For more detail, refer to https://www.medcalc.org/manual/roc-curves.php
2) To select the best scoring system for your prediction based on ROC analysis, you must perform a comparative ROC analysis. You can proceed with SAS, Medcalc, R, etc.
Performing a comparative ROC analysis on a scoring system based on the largest ROC value can interpret that a scoring system with significant p-values predicts clinical outcomes differently from other scoring systems. In addition, the overlay plot should be presented as shown below.
(Test 1~3 will be scoring systems in your study)
For Medcalc: https://www.medcalc.org/manual/comparison-of-roc-curves.php
For SAS, https://blogs.sas.com/content/iml/2018/11/14/compare-roc-curves-sas.html
R has several packages regarding to this (google it)
[Line #244~]
It is written “the inflammation-based risk scoring system showed the best performance to predict thrombosis (AUC 0.63; 95% CI: 0.52 to 0.74; P=0.020)”, but according to Table 4 its sensitivity is too low as 56.2% which means only 56.2% of thrombosis can be detected in advance. Can you say the performance is good? Its specificity is also low as 72%. Specificity more than 90% is usually recognized as a good performance. The word usage of “best performance” is not reasonable.
[Line#317]
It is written “hospitalized patients with severe COVID-19”. I'm confused that these results came from using only severe patients. Didn’t you use patients with mild or severe COVID-19?

Author Response
REVIEWER 3
This helps identify patients who need prompt treatment. But in order to generalize the author’s ideas, the validation should be conducted. The validation can be conducted with external data or internal data.
The validation with external data is not available in many cases because it is difficult to collect additional data. In that case authors can proceed with internal validation to claim the scoring system as a prediction tool. It is easily found the internal validation in Google. It is difficult to recognize the scoring system without validation results.
R= In principle, we agree with the reviewer's observation. It is best to perform an external validation of the risk scoring system to transform it into a prediction tool. However, performing this external validation would take a long time and we would now have to deal with the effects of the vaccine administration, as well as the emergence of viral variants of concern. This would involve testing the risk scoring system in a population quite different from the original one.
Regarding an internal validation, we consider that we do not have a sample size to divide it and generate a derivation cohort and a validation cohort. Furthermore, the present study is an external validation analysis of each of the risk scoring systems, since these were derived (constructed) or at least previously applied in COVID-19 patients. In support of this claim, the Obesity & Diabetes score showed a C-statistic for predicting death of 0.83 in the original study by Bello-Chavolla et al, while this figure was 0.86 in our study, confirming (externally validating) the usefulness of this clinimetric tool in COVID-19. We have added these concepts in different parts of the Discussion.
[For Table 3]
1) It is recommended to remove p-values on Table 3. Because the significant p-value obtained from ROC analysis means that AUC is significantly different from 0.5. Thus, the p-value less than 0.05 on ROC curve dose not have meaning. Instead of that, the amount of AUC value should be used. Providing the p-values from AUC can make readers confused. For more detail, refer to https://www.medcalc.org/manual/roc-curves.php
R= We agree with the reviewer. Consequently, we have already made the suggested changes.
2) To select the best scoring system for your prediction based on ROC analysis, you must perform a comparative ROC analysis. You can proceed with SAS, Medcalc, R, etc.
Performing a comparative ROC analysis on a scoring system based on the largest ROC value can interpret that a scoring system with significant p-values predicts clinical outcomes differently from other scoring systems. In addition, the overlay plot should be presented as shown below.
(Test 1~3 will be scoring systems in your study)
For Medcalc: https://www.medcalc.org/manual/comparison-of-roc-curves.php
For SAS, https://blogs.sas.com/content/iml/2018/11/14/compare-roc-curves-sas.html
R has several packages regarding to this (google it
R= We greatly appreciate this comment / suggestion. Please find the analysis you are requesting in Figure 1 (a new figure). In addition, the following paragraph was included in the Results: ”A comparative ROC analysis of the risk scoring systems that performed best in predicting each clinical outcome is presented in Figure 1. The inflammation-based risk scoring system was not significantly better (P=0.737) than the NLR in predicting vascular thrombosis. Contrarily, the Obesity & Diabetes score was better than any of the other risk scoring systems in predicting the need for mechanical ventilation (P<0.0001 for all comparisons), death (P<0.001 for all comparisons), and the composite outcome (P<0.01 for all comparisons).”
[Line #244~]
It is written “the inflammation-based risk scoring system showed the best performance to predict thrombosis (AUC 0.63; 95% CI: 0.52 to 0.74; P=0.020)”, but according to Table 4 its sensitivity is too low as 56.2% which means only 56.2% of thrombosis can be detected in advance. Can you say the performance is good? Its specificity is also low as 72%. Specificity more than 90% is usually recognized as a good performance. The word usage of “best performance” is not reasonable.
R= We agree. We have modified the text as follows: “Specifically, the inflammation-based risk scoring system showed the greatest ability to predict thrombosis (AUC, 0.63; 95% CI: 0.52 to 0.74; P=0.020), although its performance is far from optimal; while, the Obesity & Diabetes score was the best to discriminate…”. These concepts have also been modified in other parts of the manuscript.
[Line#317]
It is written “hospitalized patients with severe COVID-19”. I'm confused that these results came from using only severe patients. Didn’t you use patients with mild or severe COVID-19?
R= The wording has already been modified so there is no confusion.
The authors are indebted to the Reviewer, we are sure that their comments have substantially improved the quality of our manuscript.
Round 2
Reviewer 1 Report
The authors responded to my earlier comments. Please provide an IRB number in the section of IRB statement.
Author Response
I have already entered the IRB name and the number assigned by the IRB to the project.
I wish to thank you for your time and effort in reviewing my manuscript. I am certain that your suggestions and observations have substantially improved my paper.
Truly yours,
LA-G
Reviewer 2 Report
Thank you for your answers and the quality of this reviewing.
Author Response
Dear Reviewer,
I wish to thank you for your time and effort in reviewing my manuscript. I am certain that your suggestions and observations have substantially improved my paper.
Truly yours,
LA-G
Reviewer 3 Report
The article is well revised. Thank you for your hard work.
Author Response

(The authors gave the same response as above.)
